# Optimization of a High-Performance Poly(diallyl dimethylammonium chloride)-alumina-perfluorooctanoate Intercalated Ultrafiltration Membrane for Treating Emulsified Oily Wastewater via Response Surface Methodology Approach

**DOI:** 10.3390/membranes11120956

**Published:** 2021-12-01

**Authors:** Yusuf Olabode Raji, Mohd Hafiz Dzarfan Othman, Nik Abdul Hadi Sapiaa Md Nordin, Mohd Ridhwan Adam, Khairul Anwar Mohamad Said, Kabir Abogunde Abdulyekeen, Ahmad Fauzi Ismail, Mukhlis A. Rahman, Juhana Jaafar, Suriani Abu Bakar

**Affiliations:** 1Advanced Membrane Technology Research Centre (AMTEC), School of Chemical and Energy Engineering, Universiti Teknologi Malaysia (UTM), Johor Bahru 81310, Malaysia; yoraji@atbu.edu.ng (Y.O.R.); mohd.ridhwan@utm.my (M.R.A.); afauzi@utm.my (A.F.I.); r-mukhlis@utm.my (M.A.R.); juhana@petroleum.utm.my (J.J.); 2Department of Chemical Engineering, Abubakar Tafawa Balewa University (ATBU), Bauchi P.O. Box 0248, Nigeria; ayekeenkabir@atbu.edu.ng; 3Department of Chemical Engineering, Universiti PETRONAS (UTP), Seri Iskandar 32610, Malaysia; nahadi.sapiaa@utp.edu.my; 4Department of Chemical Engineering and Energy Sustainability, Faculty of Engineering, Universiti Malaysia Sarawak, Kota Samarahan 94300, Malaysia; mskanwar@unimas.my; 5Department of Chemical Engineering, Faculty of Engineering, University of Malaya, Kuala Lumpur 50603, Malaysia; 6Nanotechnology Research Centre, Faculty of Science and Mathematics, Universiti Pendidikan Sultan Idris, Tanjung Malim 35900, Malaysia; suriani@fsmt.upsi.edu.my

**Keywords:** response surface methodology, central composite design, emulsified oil in water, ceramic hollow fiber membrane, nanocomposite coating

## Abstract

This research aimed to investigate the ultrafiltration of water from emulsified oily wastewater through the application of surface-functionalized ceramic membrane to enhance its water permeability based on optimized parameters using a cross-flow filtration system. The interactive effects of feed concentration (10–1000 ppm), pH (4–10), and pressure (0–3 bar) on the water flux and oil rejection were investigated. Central composite design (CCD) from response surface methodology (RSM) was employed for statistical analysis, modeling, and optimization of operating conditions. The analysis of variance (ANOVA) results showed that the oil rejection and water flux models were significant with *p*-values of 0.0001 and 0.0075, respectively. In addition, good correlation coefficients of 0.997 and 0.863 were obtained for the oil rejection and water flux models, respectively. The optimum conditions for pressure, pH, and feed concentration were found to be 1.5 bar, pH 8.97, and 10 ppm, respectively with water flux and oil rejection maintained at 152 L/m^2^·h and 98.72%, respectively. Hence, the functionalized ultrafiltration ceramic membrane enables the separation efficiency of the emulsified oil in water to be achieved.

## 1. Introduction

Hazardous contaminants in the form of wastewater are primarily generated in industries. Such harmful effluents contain hydrocarbons, heavy metals, harmful microbes, and dyes [1]. Discharging the effluents into the environment requires substantial pretreatment to meet minimum regulatory standards [2]. For example, conventional approaches to the treatment and separation of oil from oily wastewater, such as coagulation, floatation, gravity settling, and ultrasonic have been found to be ineffective mainly due to low separation efficiency, process separation units of equipment being complex, and high energy cost and secondary pollution [3,4,5,6,7]. However, microfiltration (MF) and ultrafiltration (UF) processes have been adjudged as alternative approaches to treat oily wastewater effectively. This is due to the fact that the process requires low pressure to treat a high volume of effluent, and no additional chemical is required [8].

In material selection, ceramic membranes have been taken into consideration for presenting more advantages in terms of excellent solvent resistance, high permeation flux, high oil rejection, long lifetime, high thermal stability, and exhibiting chemical inertness in the area of MF and UF processes [9]. Several articles have reported various strategies of oil and water separation, especially the use of ceramic membranes and inorganic materials, for instance, alumina [10], αAl_2_O_3_ [11], kaolin [12], sugarcane bagasse waste [13], NaA zeolite [14], stainless steel [15], Cu mesh film [16], kaolin/fly ash [17], fly ash [18], magnesium bentonite [19], and ball clay [20]. Even though the membrane with MF meets the requirement for oil–water application, it is not desirable and insufficient for oil–water separation because it can easily be fouled and shorten the life span of the membrane, thereby increasing the cost of membrane maintenance and operation [21]. However, addressing these challenges of MF is the use of a UF ceramic membrane. Quite a number of literature have reported the application of UF for emulsified oily wastewater [22,23,24,25,26,27,28,29,30,31]. Oily wastewater separation processes still possess the challenge of flux decline which can be attributed to fouling of UF ceramic membranes, and thereby limits their economic viability [30]. Despite the challenges, the cost of UF is comparatively lower than conventional treatment methods of oily wastewater [32,33]. To address the impediment from the decline in the operation efficiency of UF membranes, the use of interfacial nanocomposites would improve the permeation flux and reduce the fouling of UF membranes.

The application of ceramic membranes for oily water separation gives impetus to the physical structure and surface energy of the membrane; in other words, it gives importance to the substrate with high surface roughness and suitable wettability [34]. Both influence the concept of surface wettability behavior, which usually expresses the ability of a material to get wetted when liquid encounters it. However, surface functionalization of membrane enables alteration of the membrane surface’s wetting properties for proper oil and water separation as it will exhibit surface roughness and efficiently make membrane function for oil and water separation [35]. Surface functionalization with the use of interfacial material has helped to address the challenges of fouling associated with the ceramic membrane. The preparation parameters for membranes with interfacial material fabricated by different modification approaches were found in the literature [6,36,37,38,39,40,41]. However, there has been no great attention that focuses on the simultaneous operational parameters on the performance of ceramic membrane functionalized with a nanocomposite for the application of oily water treatment. In addition, to the best of our knowledge, most of the hitherto published work focused on the use of the conventional method whereby the parameters were studied independently while other parameters were kept constant.

Response surface methodology (RSM) is a statistical approach to the multifactorial analysis of experimental design and process optimization, which offers an improved understanding of a process as compared to standard methods of experimentation since it is able to predict how the inputs affect the outputs in a complex process where different factors can interact between themselves [42]. The conventional “one-factor at a time (OFAT)” method used for optimizing a multifunctional system not only is time-consuming but often errs in the other effects between independent variables involved in the process. Moreover, this approach involves performing a number of experiments to determine the optimum conditions. The setback of the single factor optimization process can be minimized by optimizing all the affecting independent variables together with the use of central composite design (CCD) using RSM [43]. RSM involves three steps: the first step requires analysis of individual and combined parametric effects. The influence of the primary parametric variable is evaluated for process efficiency as the second step, while the third is the process optimization using RSM based regression model to achieve optimum process conditions [44]. For instance, Belgada et al. [45] studied the preparation parameters (kaolinite content loading, sintering temperature, and sintering time) of natural phosphate and kaolinite ceramic membrane using the Box–Behnken design towards textile wastewater treatment with an excellent 99% of turbidity removal and 69% of total organic carbon removal. On the other hand, Milic et al. [24] presented investigation on the process parameters influencing the ultrafiltration of oil-in-water emulsion by using the ceramic membrane-based Taguchi design approach. The optimum condition was found at pH = 7, transmembrane pressure, TMP—5 bar, and oil concentration, ϕ = 0.5 *v*/*v*%.

This present study aims to optimize the surface-functionalized ultrafiltration hollow fiber ceramic membrane for the separation of emulsified oil in water using response surface methodology based on central composite design. The number of reported studies using this approach is limited in the field of oily wastewater treatment by using hollow fiber ceramic membranes [44,46,47,48]. The functionalized surface will allow the simultaneous display of interaction between water as a polar phase and oil as a non-polar phase under the hydrophilic/underwater oleophobic approach. In other words, water molecules would be able to penetrate the surface as a result of water-induced molecular rearrangement with the hydroxyl group of kaolin membrane attached to the interface. On the other hand, oil at the interface of sodium perfluorooctanoate (PFO) and poly(diallyl dimethylammonium chloride) (PADAMAC), PDADMAC-Al_2_O_3_/PFO nanocomposite would experience low surface energy with oleophobicity because PFO has the tendencies to transform the Al_2_O_3_ surface by lowering its surface energy. The PDADMAC prevents agglomeration of alumina nanoparticles, and surface cracking of membrane surface, thereby improving the oleophobicity of the coating. The oil rejection and water flux performance tests were conducted at room temperature using a laboratory-scale cross-flow filtration setup. For accomplishing this, three independent variables are chosen, namely the feed concentration into the feed tank, the pressure of the operating system which helps to increase the quasi-steady flux, and the pH of the emulsified oil as it influences the membrane surface charges and solute adsorption mechanism.

This study continues on our previous studies on wettability improvement of the ceramic membrane by intercalating nano-Al_2_O_3_ for oil and water separation [49] and synthesis and characterization of superoleophobic fumed alumina nanocomposite coated via the sol-gel process onto ceramic-based hollow fiber membrane for oil–water separation [50]. Based upon our previous findings, the performances of the modified ceramic membrane could not produce favorable high-water flux if the oil concentration was allowed to exceed 1000 ppm. At this condition, the water flux was found to be 2.5 L/m^2^·h. However, considering multivariable parameters is highly desired to attain optimum conditions in terms of membrane performance.

## 2. Experimental Procedures

### 2.1. Raw Materials

Kaolin clay (1.97 μm) used in this research was purchased from BG Oil Chem Sdn. Bhd., Malaysia, while the alumina (1.68 µm) used in this work was obtained from Acros Denmark. The powders were stored in the oven (80 °C) for 24 h before use in order to eliminate moisture. N-methyl-2-pyrrolidone, (NMP; high-performance liquid chromatography, HPLC grade, Rathbone), polyethersulfone (PESf; Radel A-300, Ameco Performance, Cleveland, OH, USA), poly (diallyl dimethylammonium chloride, 20 wt.%; Sigma-Aldrich, Burlington, MA, USA), and sodium perfluorooctanoate (Alfa Aesar, Kandel, Germany) were used as received. All the chemicals were used without further purification.

### 2.2. Ultrafiltration Membrane Preparation

The ultrafiltration membrane was prepared by surface modification of the hollow fiber ceramic membrane. The detailed fabrication of hollow fiber ceramic (substrate) using kaolin powder was carried out by phase inversion method and sintering temperature technique from our previous study [49]. Prior to the dope suspension preparation, the kaolin powder was pre-dried in the oven at 80 °C overnight to remove the moisture. Firstly, the dispersant, Arlacel P135 was dissolved in NMP under vigorous stirring in a planetary milling jar. Once a homogenous solution was formed, the pre-dried kaolin powder was slowly added into the solution and the suspension was subsequently subjected to mechanical stirring (NQM-2 planetary ball mill) at 194 rpm for 48 h to ensure proper dispersion of kaolin powder. After 48 h of milling, PESf was added in the ratio of 1:8 for ceramic content loading and milled for 48 h to achieve proper dispersion and binding of the mixture. Then the dope was degassed under the vacuum, and then gently stirred for 30 min to ensure that any air bubbles trapped were entirely removed from the dope. In addition, kaolin-based hollow fiber ceramic precursor was subjected to sintering temperatures in the tubular furnace (XY-1700 MAGNA) at a heating rate of 3 °C/min for 2 h. The temperature program rose from the initial room temperature, and with plateaus at 250 °C for the elimination of dispersing agent and 650 °C for other additives such as the binder and solvent. Finally, the furnace temperature dropped to room temperature. The choice of the sintering temperatures was taken based on the previous studies [49].

The top coating layer for the kaolin-based hollow fiber ceramic membrane was prepared using the sol-gel method from fumed Al_2_O_3_ and polyelectrolyte-fluorosurfactant and deposited using a simple dip-coating method based on previous studies [50]. Poly (diallyl dimethylammonium chloride) was initially diluted to 1.0 mg/mL in 10 mL of deionized water, followed by the addition of Al_2_O_3_ (0.4 g) which was ultrasonically dispersed into the solution. Then, it was magnetically stirred for 2 h to absorb PDADMAC on the Al_2_O_3_ surface. Finally, sodium perfluorooctanoate (20 mL, 0.1 M) was added dropwise under stirring, making PFO anions coordinate to quaternary ammonium groups of PDADMAC. The mixture was stirred for 2 h and then placed in the ultrasonic process for 30 min and stirred continuously. The final products, PDADMAC-Al_2_O_3_/PFO (PAP-BM), were obtained by filtrating, rinsing, and drying.

The dip-coating process was conducted; firstly, the membrane surface was abrased and then washed in a mixture of deionized water and ethanol (1:2) to enable hydroxyl group formation on the bare membrane surface. Both ends of hollow fiber ceramic membranes of about 10 cm in length were potted with PTFE film tape before vertical dip-coating of the membrane was conducted. Upon retraction, the membrane was dried for 24 h at room temperature. PAP as the synthesis products (0.4 g) solutions was dispersed in ethanol (20 mL), respectively under sonication. The resulting suspensions were dip-coated onto the substrate. The PAP-based coatings were dried at room temperature for 1 h allowing the ethanol to evaporate completely. The dip-coating conditions for microfiltration layer coating where the supporting dipping time was 10 s and the support withdrawal time limit of 5 s were applied. The process was continuously repeated after each successful drying time of 30 min. While the ultrafiltration characteristic was introduced by dip-coating deposition of alumina nanocomposites suspension synthesized using the sol-gel approach as reported in our previous studies [50]. Figure 1 shows the surface morphologies, topological image, and pore size distribution of the modified ceramic membrane. PAP-BM pore size distribution demonstrated a mode shift from microfiltration to PAP-BM ultrafiltration. In addition to MIP analysis, the presence of a UF layer on the substrate can also be confirmed by SEM. An average pore size of the UF layer of 0.053 µm was obtained for PAP-BM. This is in agreement with the reported literature on the ceramic UF membrane pore size ranging from 0.001 to 0.1 µm being most suitable for the separation of macromolecules and small molecules of emulsified oily wastewater.

### 2.3. Experimental Design

#### 2.3.1. Filtration Experiment

The experimental steps for the oil–water separation were conducted on the ultrafiltration system using a modified membrane. Figure 2 shows the experimental set-up for the oil–water separation under a cross-flow ultrafiltration system with a 9″ (22.86 cm) stainless-steel module that is 0.5″ (1.27 cm) in diameter. The cross-sectional area of the membrane stainless-steel module was 100.30 cm^2^, and the membrane surface area in contact with fluid was 4.58 cm^2^. The set-up also includes valves, pressure gauge (with max. input: 15 bar), booster pump with normal flow rate (1.0 L/min) and maximum inlet pressure (about 15 bar), connecting tubes, and feed tank. The feed tank was stainless steel, with a capacity of 5 L. The inlet pressure and outlet pressure were controlled by the by-pass and outlet valves (valve 1 and valve 2). Prior to the oily wastewater experiments, emulsified oily water was prepared by mixing red palm oil and water using sodium dodecyl sulfate (SDS) as the emulsifying (surfactant) agent. The weight ratio of SDS to oil was 1:9 and added to the feed tank at different experimental conditions: concentrations, pressure, and pH based on the design of the experiment using central composite design. Twenty experimental runs were obtained for the performance tests. The following operating parameters, pressure, oil concentration, and pH, were carefully controlled. The emulsified oil–water was fed via feed pump (max. output: 3 bar) with a booster pump to be transferred at a nominal flow rate of oily wastewater from a feed tank through a cross-filtration of the membrane module, then through the potted side of the adapter. The system allows feed permeation across and out from the lumen side of the membrane and cycles back (retentate) into the feed tank. For analyzing the permeate, the permeate sample was collected after 20 min of stability for 90 min. The liquid detergent washing loop was used after each oily wastewater filtration. The membrane permeation flux was measured by collecting the permeate volume in a graduated cylinder. The experimental set-up including piping, pump, feed tank, etc., was entirely washed to be used for the next run. The oil rejection, R and water flux, $J_{i}$ were determined using Equations (1) and (2), respectively
(1)$$R\;\left(\%\right)=\frac{C_{f}-C_{p}}{C_{f}}$$
(2)$$J_{i}=\frac{V}{A\times t}$$
where $C_{f}\;\mathrm{and}\;C_{p}$ represent the concentration of the feed and permeate, respectively. The absorbance of the prepared oily water was measured using UV/vis spectrophotometer (PerkinElmer Lambda 25) at the wavelength taken between 200 and 450 nm as a calibration curve. The absorbance, A (L·mol^−1^cm^−1^) was determined using the Beer–Lambert equation as reported by [19]. $J_{i}$ denotes the representation of Jo for the oil and Jw for the water flux (L/h·m^2^), V is the volume of the water permeated through the membrane (L), A is the surface area of hollow fiber membrane (m^2^), and Δt is the total permeation time (h).

#### 2.3.2. Analyses

In the interest of giving a definite description of the factors, the dependent and independent variables involved in the experimental design were denoted as indicated in Table 1. The independent variables (levels), lower and upper limits used in the experimental design are represented in Table 1.

After designing and inputting the levels for the independent variables into Design-Expert 7.0.0. According to the CCD method, 20 runs (8 cube points, 6 center points in cube, and 6 axial points) were generated without the alpha values in the design. However, the experimental error determined using Equations (3) and (4) is the second-order polynomial model of the experimental design. The percentage error was computed using Equation (5). The mathematical model that explains the relationship between responses (dependents) and independent variables is as follows:(3)$$\sigma \;\left(\%\right)=\frac{Y_{AEV}-Y_{REV}}{Y_{REV}}\times 100$$
where σ is the experimental error, $Y_{AEV}$ is the response for actual experimental of oil rejection and water flux and $Y_{REV}$ is the response for revalidated experimental of oil rejection and water flux.
(4)$$Y_{r,f}=A_{O}+\overset{k}{\underset{i=1}{\sum }}A_{i}B_{i}+\overset{k}{\underset{i=1}{\sum }}A_{ij}B_{i}^{2}+\overset{k}{\underset{i,j=1,i\neq j}{\sum }}A_{ij}B_{i}B_{j}+C$$
where $Y_{r,f}$ = corresponding responses (oil rejection and water flux), $B_{i}$ = input variables (feed concentration, pressure, and pH), $B_{ii}$ = square term of input variable, and $A_{O}$, $A_{i}$, $A_{ij}$, $A_{ii}$ are the unknown regression coefficients, C = model error.
(5)$$\%\;error=\left(\frac{Y_{i}-Y_{j}}{Y_{i}}\right)\times 100$$
where $Y_{i}$ is the actual value of oil rejection and water flux and $Y_{j}$ is the predicted value of the oil rejection and water flux.

#### 2.3.3. Membrane Antifouling Evaluation

An experiment with model foulant (red palm oil with average particle size distribution 56.8–77.23 µm) was performed to estimate the antifouling property of the membrane samples. The concentrations of foulant solutions were 10 ppm, 5000 ppm, and 1000 ppm. Prior to the antifouling experiment, the pure water flux, $J_{W,1}$ (L/m^2^·h) of the uncoated ceramic membrane was conducted at 2 bars, and then the feedstock tank was filled with foulants solution. A cross-flow ultrafiltration test was performed under the same pressure; the permeate was collected at every 20 min for 2 h, a steady flux, $J_{p}$ (L/m^2^·h) for the foulant was obtained over a period of 2 h. After the 2 h filtration process, the membrane was cleaned with RO water and another pure water flux, $J_{w,2}$ (L/m^2^·h) was conducted for 2 h at 2 bars as the recovery flux. The recovery flux percentage (RFP), reversible fouling (RF) resistance percentage, and irreversible fouling (IF) resistance percentage were defined using Equations (6)–(8), whereas, the total fouling (TF) resistance was used to determine the degree of total flux loss caused during fouling process of the PAP-BM membrane, using the following equations: Shen et al. [51]; Vatanpour et al. [52]. However, higher RFP indicates a better antifouling property of the membranes. The fouling resistance for the chemical processing stream (vegetable oils) is 0.0005 as suggested in [53]. Hence, the resistance formed during the filtration process can point to the fouling of the membrane.
(6)$$\mathrm{RFP}\;\left(\%\right)=\left(\frac{J_{w,2}}{J_{W,1}}\right)\times 100$$
(7)$$\mathrm{RF}\;\left(\%\right)=\left(\frac{J_{w,2}-J_{O}}{J_{W,1}}\right)\times 100$$
(8)$$\mathrm{IF}\;\left(\%\right)=\left(\frac{J_{w,1}-J_{w,2}}{J_{w,1}}\right)\times 100$$
(9)$$\mathrm{TF}\;\left(\%\right)=\left(\frac{J_{w,1}-J_{P}}{J_{w,1}}\right)\times 100$$

## 3. Results and Discussion

### 3.1. Experimental Design Using Response Surface Methodology

The probable mechanism of hydrophilic/oleophobic behavior of the coating is primarily related to the micro-nano hierarchical structure of the coating due to enhanced roughness caused by the presence of Al_2_O_3_ nanoparticles and their effective intercalating into hydrophilic polymer PDADMAC and using kaolin-based substrate plummeting the surface tension due to the presence of PFO, robust attachment of the film to the substrate. Eventually, excellent oil–water separation efficiency and improved water flux can be attained (Figure 3).

Apart from the physical and chemical properties of the modified ceramic fiber membrane, there are external factors that intensely influence the emulsified oil–water separation and the oil rejection performance of the membrane. Factors such as feed concentration, feed pH, and pressure have a significant effect on the modified membrane in terms of membrane water flux and oil rejection.

Table 2 provides the results from the experimental design using RSM. From the results, the maximum oil rejection (*Y_AV_* at run no.8) obtained was 99.70% when the values of feed concentration, pressure, and pH were 10 ppm, 0 bar, and pH 4, respectively. Moreover, the minimum oil rejection (*Y_AV_* at run no.1) of 40.0% was found at 10,000 ppm, 1.5 bar, and pH 7. It can be noted from the results that the increase in the feed concentration from 5000 to 10,000 ppm (run 9 to 12) with the increase in pressure and pH indicates a decline in the oil rejection. This can be attributed to the multivariable nature of the design, even though one of the variables was constant. For instance, the interactive effect of pressure at runs 9, 10, 16 when factors the feed concentration and pH values remained the same, there was an increase in oil rejection from 55.00 to 90.69% while water flux showed a decline from 160.0 to 64.00 L/m^2^·h. The effect of increased pressure caused compression of cake or gel layer and resulted in the flux decline. This contrasts with the cases for pH at runs 17, 18, and 19 where feed concentration and pressure values were the same, the findings suggest that both the oil rejection and water flux increased from 49.65 to 85.00% and from 64.00 to 112 L/m^2^·h, respectively. This can be attributed to the role played by pH to influence the membrane surface charges and oil–water separation mechanism [22]. However, the predicted oil rejection values revealed a slight difference of ±2.00 compared with actual values. Similarly, the maximum water flux (*Z_AV_* at run no.6) obtained was 176.00 L/m^2^·h when the values of feed concentration, pressure, and pH were 10 ppm, 3.0 bar, and 4 respectively. Moreover, the minimum water flux (*Z_AV_* at run no.17) of 56.00 L/m^2^·h was found at 10,000 ppm, 0 bar, and pH 10. However, the predicted values of water flux revealed a wider margin of +5.00 when compared with predicted values. Therefore, the trend can be said to be directly or indirectly proportional to each of the independent variables, obviously due to the concentration gradient and thickness of the feed solution. The water could be extracted and more fouling can be promoted, and vice versa. It also favors lower feed concentration and pressure. The significance of this is that multivariable helps in the experimental design and optimization of many independent and dependent variables.

### 3.2. Oil Rejection Response

From the results (Table 3), it was found that the p-values of factors AB and AC were greater than the significance level of 0.05. This high p-value of AB and AC can be ascribed to the hierarchical problem of the model and that these two independent variables had no individual effect on the oil rejection. The analysis of variance (ANOVA) revealed that this model was significant with a p-value of 0.0001. The overall model was found to be significant with “lack of fit” not significant, F-value of 0.52 (that is, there are 68.42% chances that “lack of fit F-value” this large could occur due to noise), and the R-squared value of 0.9921 confirmed their agreement with the experimental data. Furthermore, R-squared evaluates the discrepancy or variance in the apparent values, which could be explained by the independent variables and their interactions over the design of the specific factors. This R-squared of 0.9921 indicates that the response variation of 99.21% of the total variation could be described by the model, and only 0.79% of it was not described by the model. Therefore, the model equation was better in representing the oil rejection concerning the three independent variables.

Moreover, the ANOVA of the modified cubic regression model confirmed that the model was considered significant (*p* < 0.05). The linear model terms of feed concentration (A), pressure (B), and pH (C) and the quadratic model of denoted terms *A*^2^, *B*^2^, *C*^2^, *A*^2^*B*, *A*^2^*C*, and *A*^2^*B*^2^ were significant (*p* < 0.05), indicating that the three independent variables had a distinct effect on the oil rejection. Furthermore, Figure 4 suggests that the experimental results of oil rejection value were close to the predicted value.
(10)$$Y_{oil\;rejection}=85.70-26.68\times A-17.85\times B+17.68\times C$$
(11)$$\begin{array}{c} -0.90\times AB-0.60\times AC-13.84\times A^{2}-12.85\times B^{2}-18.37\times C^{2}\\ +16.05\times A^{2}B-17.20\times A^{2}C+41.45A^{2}B^{2} \end{array}$$

### 3.3. Water Flux Response

From the results (Table 4), it was found that *p*-values of factors *C*, *AB*, *BC*, *A*^2^, *B*^2^, and *A*^2^*B* were greater than the significance level of 0.05. This high *p*-value can be ascribed to the hierarchical problem of the model. The analysis of variance (ANOVA) revealed that this model was significant with a *p*-value of 0.0075 (see also Table 4). The overall model was found to be significant with insignificant lack of fit; F-value of 0.94 (that is, there are 50.91% chances that “lack of fit F-value” this large could occur due to noise), and the R-squared value of 0.8635 confirmed their agreement with experimental data. Additionally, it measures the variability in the observed response values, which can be described by the independent factors and their interactions over the range of the corresponding factors. This R-squared indicates that the response variation of 86.35% of the total variation could be described by the model, and only 13.65% of it was not described by the model. Hence, the model equation was better in representing the water flux with respect to the three independent variables.

Furthermore, the ANOVA of the modified cubic regression model proved that the model was significant (*p* < 0.05). The linear model terms of feed concentration (A) and pressure (B), were significant (*p* < 0.05), signifying that these three independent variables had a discrete effect on the oil rejection. On the contrary, pH (C) and the quadratic model of denoted terms *A*^2^, *B*^2^, *C*^2^, *A*^2^*B*, *A*^2^*C*, and *A*^2^*B*^2^ were insignificant (*p* > 0.05), implicating that one independent variable (C) did not affect the water flux. Based on their p-values, it can be concluded that the feed concentration has a greater effect on the oil rejection compared to pH. Furthermore, Figure 5 suggests that the experimental data value water flux value was far from the predicted value.
(12)$$\begin{array}{ll} Y_{oilflux}=98.10 & -44.00\times A+48.00\times B+5.60\times C-4.00\times AB-4.00\times AB+5.50\times B^{2}+9.50\times B^{2}\\ & -36.00\times A^{2}B \end{array}$$

### 3.4. Response Surface of Contour and 3D Plots on Separation Efficiency: Interaction Effects between the Responses and Primary Process Variables

The interactive effects with an optimum oil rejection and water flux from feed concentration and pH at constant pressure are shown in Figure 6. The contours and 3D plots provide the interactive effects with an optimum oil rejection and water flux from feed concentration and pressure at constant pH are shown in Figure 7. In addition, the pressure and pH effects at constant feed concentration are shown in Figure 8.

According to the contour plot in Figure 6a(i), it can be observed that variation of feed concentration and pressure had a significant effect on oil rejection; the variations of feed concentration from 7502.50 to 10,000 ppm and pressure from 2.25 to 3 bars led to the decrease in oil rejection. However, this decrease is more significant at the highest level of the feed concentration (10,000 ppm) and pressure (3 bar) compared to the lower-level interaction between feed concentration (10 ppm) and pressure (1 bar). This variation can be ascribed to the high feed concentration interfaced with the coated layer of the membrane at high pressure, thereby it accumulated in the form of the cake layer and resulted in the reduction in the oil rejection from 69.32% to less than 57.20%. This observation is consistent with [54]. Additionally, when only one independent parameter was varied, less impact was observed. As seen in the 3D surface graphs of the results, the maximum oil rejection (97.65%) was achieved within the design points by varying the conditions at low pressure and low feed concentration (Figure 6a(ii)).

The contour plot as indicated in Figure 6b(i) showed a further decline in the water flux from 91.48 to 70.73 L/m^2^·h when the feed concentration increased from 6253.75 to 10,000 ppm and much lower pressure from 0.37 to 0.0 bar. In most of the studies on UF membranes, the operating pressure for optimum performance is kept between 2 and 10 bar. However, in this study, at nearly 0 bar and extreme feed concentration, the water flux was low. First, the reason for this is that the flip-flop mechanism of oleophobic-hydrophilic for oil–water separation suggests the penetration of water through the membrane surface could be a time-dependent process [39]. The implication of this trend in the performance of the membrane is the cost of operation as it can lead to low fouling resistance in the membrane surface. The 3D plot in Figure 8b(ii) reveals an interactive effect of independent variable such that water flux was highest at lower feed concentration and higher operating pressure at a constant pH of 9. The figure showed a decline in water flux at high feed concentration and pH at a pressure of 3 bars. This indicates a less interactive effect of high feed concentration and lower pH values, which is less appreciable. It is, however, preferable to maintain low pressure but high flux as it helps reduce the operating cost and a system operating at low pressure usually lasts longer and would require much maintenance compared to high-pressure equipment.

The results suggest that oil rejection decreases with an increase in feed concentration and a decrease in the pressure at a constant pH 9. The interaction between pressure and feed concentration at a constant pH illustrated that feed concentration had an immense effect on oil rejection. Theoretically, the emulsified oil aggregates influence the pore size distribution of the membrane surface. In other words, as the concentration of oil in the emulsion increased, smaller oil droplets will clump to form larger droplets over time and at higher pressure, with these larger droplets causing a blockage in the membrane pores [22].

Similarly, as seen in Figure 7a(i),a(ii), the variations of feed concentration (from 10 to 10,000 ppm) and pH (from 4 to 10) led to a decline in oil rejection. The decline was apparent between feed concentration from ~3250 to ~9000 ppm and pH of 4.0–5.0 based on the contour with oil rejection of 38.36%. The adjacent contour shows an increase in oil rejection at 46.51% which was contributed to an increase in pH from 5.0 to 7.0 while the feed concentration spanned from 2500 to 10,000 ppm. Hence, from this observation, to design an oil separation system that does not tolerate extreme pH, a system with 6.5–7.0 that could maintain 50.44% oil rejection should adopt a feed concentration between 5005 to 7000 ppm. However, this decline is more significant at the highest level of the feed concentration (2500 ppm), and pH (below 5 and above 8) compared to the interactive effect at the center, at pH 7 and feed concentration (10 ppm). Figure 7b(ii) reveals an increase in oil rejection when feed concentration decreased and favored the pH between 4 and 10 at a constant pressure of 3 bars. As seen in the 3D surface graphs of the results, the maximum oil rejection (99.70%) was achieved within the design points by varying the pressure and feed concentrations (Figure 7b). This is implicative of the emulsified medium’s chemical stability when the optimum oil rejection was obtained at pH 7 [54].

Figure 7b also illustrates the variation for the interaction effect of the feed concentration and pH on water flux when the pH was kept constant. The variation from the interaction effect of feed concentration (from 10 to 1000 ppm) and pH (from 4.0 to 10.0), resulted in a decline in the water flux from 136.67 to 70.53 L/m^2^·h. The significance of this decrease can be noted at the highest level of feed concentration and the lowest level of pressure of the contour plot. The highest water flux was achieved at ~154 L/m^2^·h, at the lowest feed concentration and higher pressure. This can be attributed to the presence of nanoparticles in the membrane, which enhances the water permeability through the hierarchical rough surface of the membrane [55]. Figure 9b(ii) reveals an interactive effect of an independent variable such that water flux was highest at lower feed concentration compared to high feed concentration and pH between 5 and 10 at a constant pressure at 3 bars.

Additionally, Figure 8 shows the interaction effect of pressure and pH at constant feed concentration on the water flux. The interaction between pressure and pH was observed, as seen in the contour plot in Figure 8a. An increase in pressure from 0 to 3 bar and at the same time the pH (from 4 to 10) revealed a simultaneous increase in the water flux. The effects are more pronounced at higher interactive factors as this favors the water flux. The highest water flux was obtained at 155.60 L/m^2^·h when feed concentration was kept at 5005 ppm. The results reflect Darcy’s law relating the proportionality of pressure and permeate flux while the pH influences the water absorption into the membrane surface. The 3D plot in Figure 8b shows a decline in water flux at lower pressure and at both lower and higher pH values when the feed concentration is kept constant at 5005 ppm. It is very important to note that the contours showed distinct intersections between pH and pressure as the water flux increases from 121.467 to 139.33 L/m^2^·h. This indicates that the interactive effect of pressure and pH significantly improves the water flux.

Therefore, as explained earlier, oil rejection and water flux had shown a significant response towards the operation parameters; hence, the optimum condition based on maximum oil rejection was obtained at 98.73.7% and water flux at 152 L/m^2^·h at a feed concentration of 10 ppm and pressure of 1.5 bar at constant pH of 7.

### 3.5. Optimization Desirability

The optimum values of various parameters are plotted in a 3D surface graph and are shown in Figure 9. The figure reveals the optimum level of the process parameter. The highest desirability value is close to unity, and all values lie between 0.5 and 1, which means that the values are close to the ideal values of desirability. The desirability of the response surface methodology technique was adequate in predicting the optimum solution of input parameters as follows: feed concentration 152 L/m^2^·h, pressure 1.5 bar, and pH 9.98. The significance of these findings suggests that pressure at 1.5 bar improves the water flux and pH 9.98 enhances the membrane surface energy charges and solute adsorption mechanism.

### 3.6. Confirmatory Test

Experimental validation of optimum conditions was carried out in order to revalidate the results obtained at optimized conditions. The ultrafiltration experiment using a membrane with three repetitions was compared with the results obtained from the model, as tabulated in Table 5. The results depict the predicted values from the model and average experimental values as obtained at optimum conditions. The experiment values for oil rejection and water flux were in good agreement with the predicted value. The average experimental value for oil rejection was 98.86 ± 1% and was satisfactorily close to the predicted value of 98.72% with a deviation of 0.14%. Furthermore, for water flux, the experimental value was 151.3 L/m^2^·h at 1.5 bar, and was close to the predicted value of 152 L/m^2^·h at 1.5 bars. Hence, the predicted model was found to be appropriate in predicting the oil rejection and water flux for the membrane. Based on Table 6, about ~57% of the previous works applied BBD for the optimization process of ceramic membranes. On the other hand, there are only a few studies on the CCD method for the optimization of UF membrane. This can be ascribed to a higher desirability efficiency of BBD as compared to CCD, because it is less expensive to run with the same number of factors. However, CCD provides higher accuracy if there are a missing number of runs which may not critically dependent on the model as compared to BBD.

The interfacial coated material such as the fluorinated alkyl group played a significant role in oil rejection. It helps to lower surface energy while the nanoparticle enhances the physical surface structure since it eliminates fouling related to cake formation. The characteristic of the membrane PAP-BM was further examined for antifouling properties with emulsified oil feed concentrations (10 ppm, 5000 ppm, and 10,000 ppm). The water flux recovery percentage (FRP), reversible fouling (RF), irreversible fouling (IF), and total fouling (TF) percentage were determined. Figure 10a shows the total fouling (TF), reversible fouling (RF), and irreversible fouling (IF) for each of the three prepared membranes. As revealed in Figure 10a, PAP-BM has the resistance TF of 73.45% efficiency. For the resistance IF efficiency, PAP-BM showed 14.09% and 52.0% for resistance RF efficiency. Figure 10b reveals the fouling resistance of PAP-BM. The results suggest that the foulants can easily be removed due to the enhanced membrane surface topology and the presence of interfacial materials, which helps in oil repellence. The fouling interaction between membrane and foulant can be classified as either reversible or irreversible. Reversible fouling suggests that there is less interaction, causing the substrate to be easily backwashed in water while irreversible fouling indicates high interaction between the substrate and foulant as it poses challenges of backwashing and a drawback in the lifespan of the membrane. Figure 10b represents the FRP (%) for the membrane. PAP-BM with 10 ppm showed 83.4% water recovery. The recovery for PAP-BM with 10,000 ppm was 51.0%. High-water flux recovery obtained can be attributed to the presence of alumina nanocomposites, which improve the hydrophilicity or the modified substrate. Hence, the performances showed that the PAP-BM exhibited the best antifouling property compared to the PP-BM and BM. Hence, the performances showed that the PAP-BM exhibited the antifouling property.

## 4. Conclusions

From the results obtained in this study, it has been revealed that CCD of RSM has been successfully useful in the treatment of oily wastewater. The oil rejection and water flux were found to be largely affected by the feed concentration, pressure, and pH. Additionally, the operating pressure and feed concentration were found to influence the system responses (the oil rejection and water flux) significantly. The results of the ANOVA showed that the oil rejection and water flux models were significant with *p*-values of 0.0001 and 0.0075, respectively. In addition, good correlation coefficients of 0.997 and 0.863 were obtained for the oil rejection and water flux models respectively. The optimum conditions of the oily wastewater separation were obtained to be 10 ppm, 1.5 bar, and 8.97 for the feed concentration, pressure, and pH, respectively. Under these conditions, 152 L/m^2^·h of water flux and 98.72% of oil rejection were achieved experimentally.

## Figures and Tables

**Figure 1 membranes-11-00956-f001:**
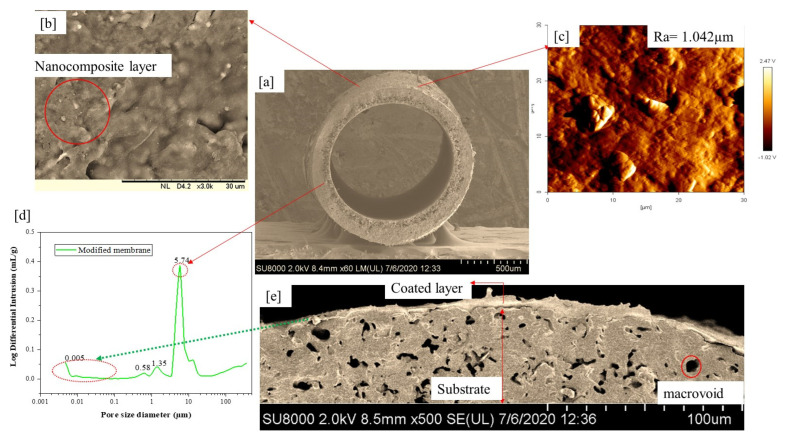
(**a**,**b**,**e**). FESEM cross-sectional image, (**b**) outer layer surface morphological images; (**c**) of the surface roughness topological image; (**d**) pore size distribution of the coaled layer of the hollow fiber ceramic membrane. Adapted from Ref. [50].

**Figure 2 membranes-11-00956-f002:**
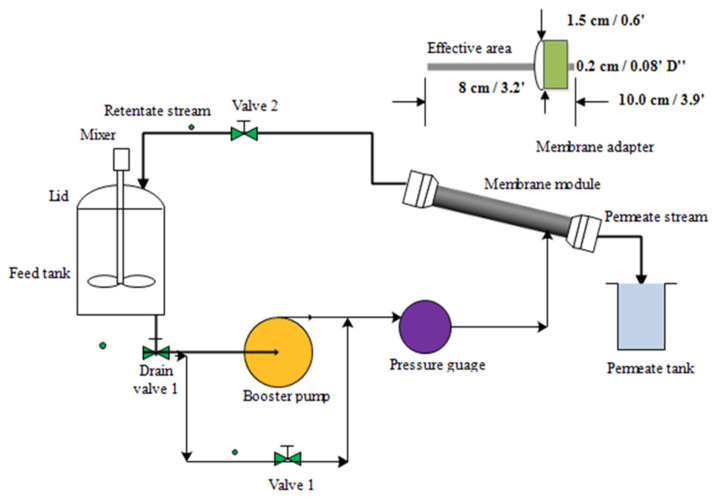
Schematic representation of cross-flow ultrafiltration system of a modified membrane with membrane adapter.

**Figure 3 membranes-11-00956-f003:**
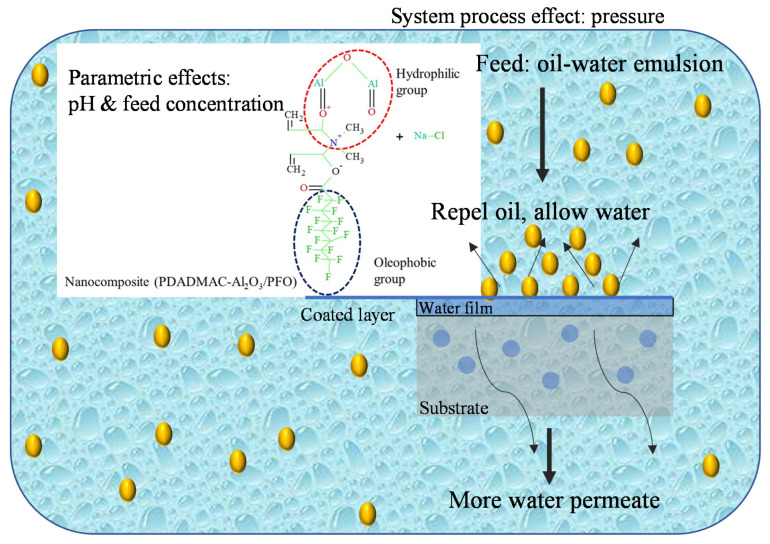
Mechanism separation of ultrafiltration ceramic membrane for the emulsified oily water.

**Figure 4 membranes-11-00956-f004:**
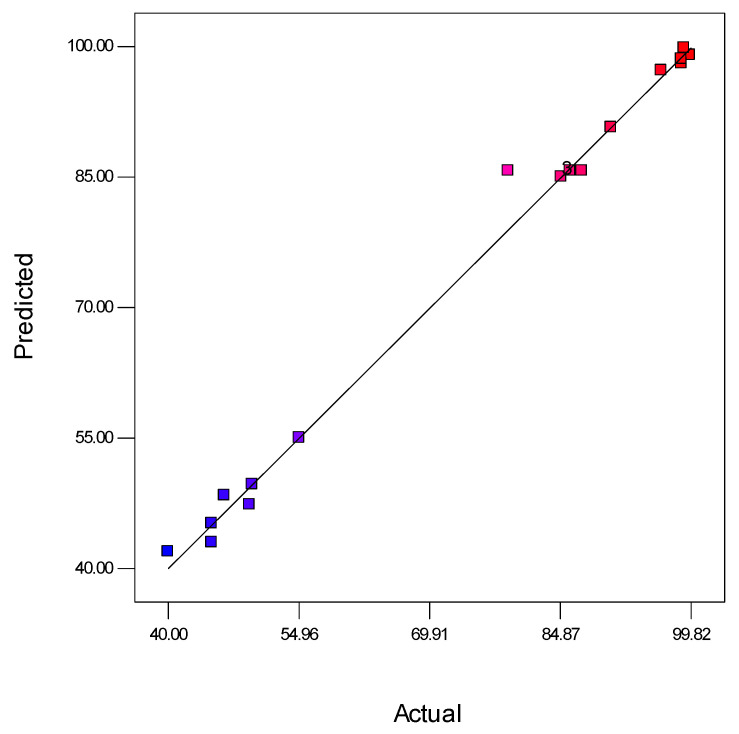
Analysis of predicted and actual response for oil rejection (%). Blue dots indicate <50% favours lower interaction between predicted and actual values of oil rejection and red dots indicate >80% favours higher interaction between predicted and actual values of oil rejection.

**Figure 5 membranes-11-00956-f005:**
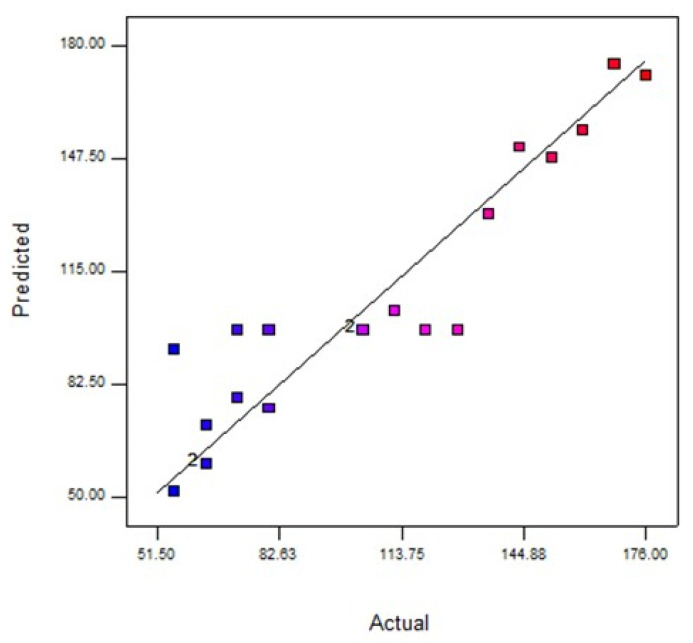
Analysis of predicted and actual response for water flux (L/m^2^·h). Blue dots indicate interaction favours predicted values of water flux and red dots indicate interaction favours actual values of water flux.

**Figure 6 membranes-11-00956-f006:**
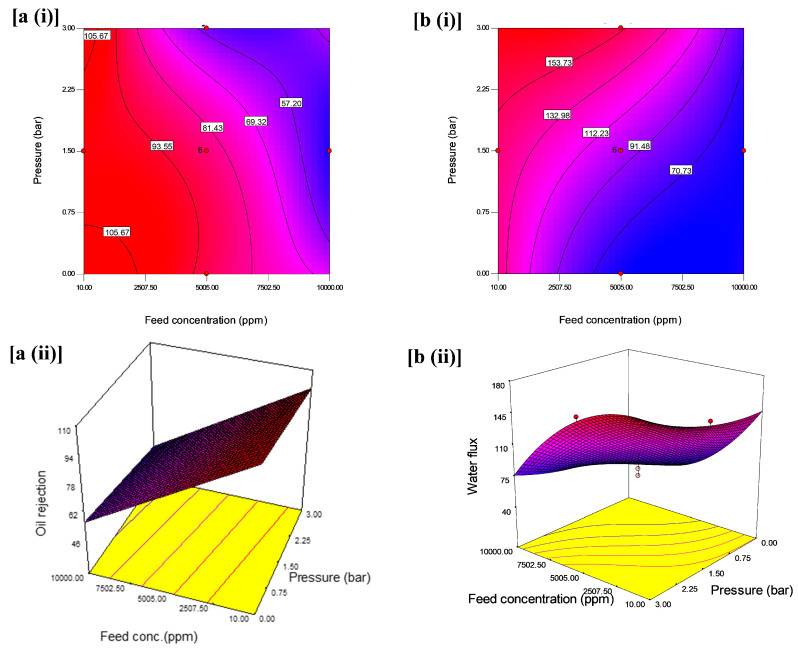
(**i**) Contour plot and (**ii**) 3D surface graph: (**a**) for the interaction effect of feed concentration and pressure on oil rejection (%) and (**b**) for the interaction effect of feed concentration and pressure on water flux (L/m^2^·h).

**Figure 7 membranes-11-00956-f007:**
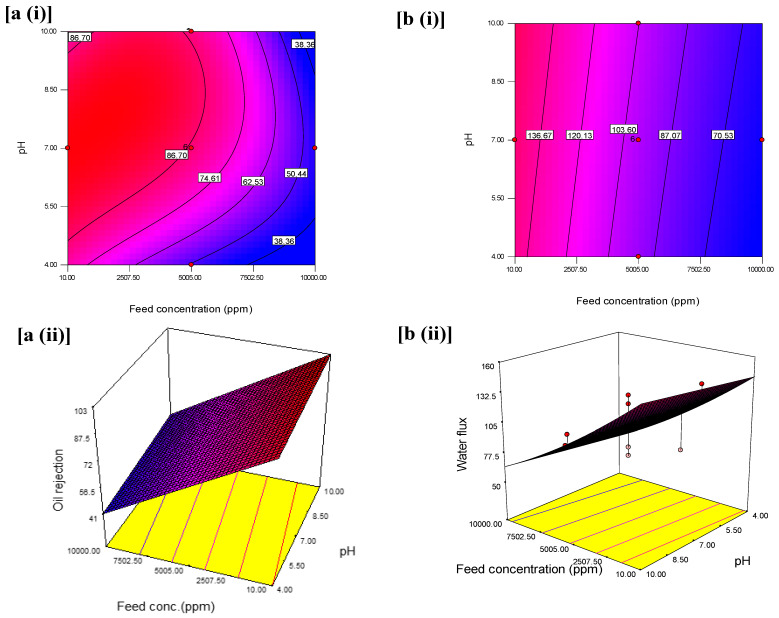
(**i**) Contour plot and (**ii**) 3D surface graph (**a**) for the interaction effect of feed concentration and pH on oil rejection (%) and (**b**) for the interaction effect of feed concentration and pH on water flux (L/m^2^·h).

**Figure 8 membranes-11-00956-f008:**
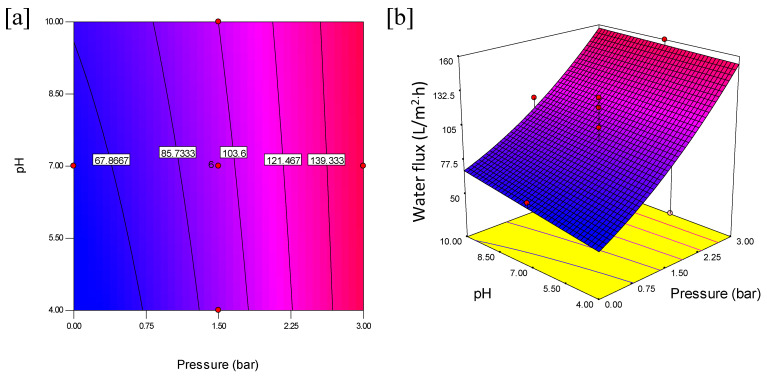
Contour plot (**a**) and 3D surface graph (**b**) for the interaction effect of pressure and pH on water flux (L/m^2^·h).

**Figure 9 membranes-11-00956-f009:**
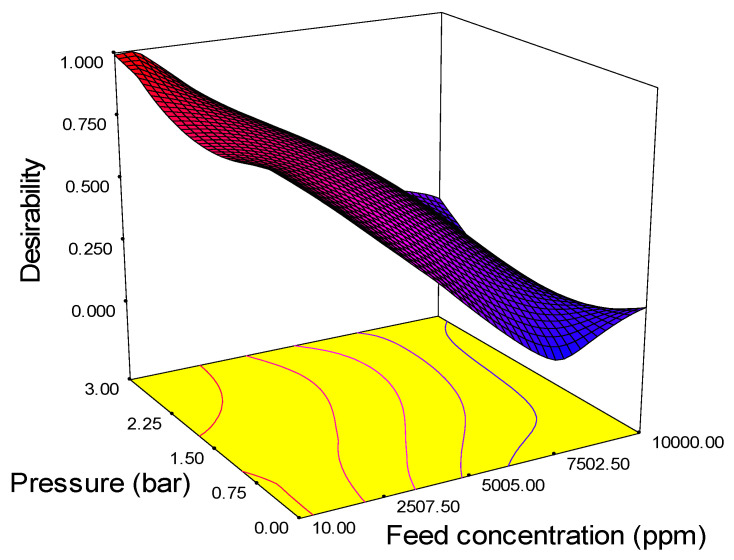
3D surface graph and contour plot on desirability based on the interaction effect of pressure and feed concentration.

**Figure 10 membranes-11-00956-f010:**
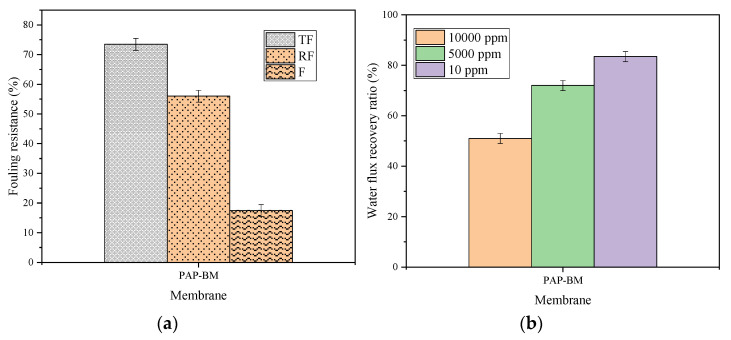
PAP-BM: (**a**) Fouling resistance efficiency and (**b**) water flux recovery percentage.

**Table 1 membranes-11-00956-t001:** Notation of the factor, levels of independent variables, and responses of the dependent variable in the experimental design.

S/N	Notation	Factor	Response	Unit	Lower Limit	Upper Limit
1	A	Feed concentration		ppm	10	10,000
2	B	Inlet pressure		bar	0	3
3	C	pH			4	10
4	Y		Oil rejection	%		
5	Z		Water flux	L/m^2^·h		

**Table 2 membranes-11-00956-t002:** Independent and process (dependent) variables of the experimental.

Run No.	A	B	C	$Y_{AV}$	$Y_{PV}$	Y	$Z_{AV}$	$Z_{PV}$	Z
1	10,000	3.00	4	40.00	41.92	−4.80	80.00	75.50	5.63
2	10,000	0.00	10	46.46	48.38	−4.13	56.00	70.70	−26.25
3	10,000	3.00	10	45.00	42.99	4.47	72.00	78.70	−9.31
4	10	0.00	10	99.03	99.82	−0.80	144.00	150.70	−4.65
5	10	3.00	10	98.75	98.05	0.71	168.00	174.70	−3.99
6	10	3.00	4	96.43	97.22	−0.82	176.00	171.50	2.56
7	10	1.50	7	98.72	98.54	0.18	152.00	147.60	2.89
8	10	0.00	4	99.70	99.00	0.70	136.00	131.50	3.31
9	5005	3.00	7	55.00	55.00	0.00	160.00	155.60	2.75
10	5005	1.50	7	87.22	85.70	1.74	128.00	98.10	23.36
11	10,000	0.00	4	49.33	47.32	4.07	56.00	51.50	8.03
12	10,000	1.50	7	45.00	45.18	−0.40	64.00	59.60	6.88
13	5005	1.50	7	78.92	85.70	−8.59	104.00	98.10	5.67
14	5005	1.50	7	86.00	85.70	0.35	80.00	98.10	−22.65
15	5005	1.50	7	87.35	85.70	1.89	72.00	98.10	−36.25
16	5005	0.00	7	90.69	90.69	0.00	64.00	59.60	6.88
17	5005	1.50	4	49.65	49.65	0.00	64.00	98.10	−53.28
18	5005	1.50	7	87.35	85.70	1.89	120.00	98.10	18.25
19	5005	1.50	10	85.00	85.00	0.00	112.00	98.10	12.41
20	5005	1.50	7	87.35	85.35	2.29	104.00	98.10	5.67

$Y_{AV}$ is the actual value of oil rejection, $Y_{PV}$ is the predicted value of oil rejection, $Z_{AV}$ is the actual value of water flux and $Z_{PV}$ is the predicted value of water flux. % error between the actual and predicted value of the oil rejection and water flux.

**Table 3 membranes-11-00956-t003:** Analysis of variance of the developed oil rejection-based modified cubic model.

Source	Sum of Squares	Df	Mean Square	F-Value	*p*-Value
Model	9321.30	11	847.39	91.20	<0.0001 (significant)
A-Feed concentration	7120.36	1	7120.36	766.32	<0.0001
B-Pressure	636.81	1	636.81	68.54	<0.0001
C-pH	624.81	1	624.81	67.24	<0.0001
*AB*	6.55	1	6.55	0.71	0.4254
*AC*	0.03	1	0.03	0.00	0.9570
*A* ^2^	287.25	1	287.25	30.91	0.0005
*B* ^2^	247.81	1	247.81	26.67	0.0009
*C* ^2^	506.37	1	506.37	54.50	<0.0001
*A* ^2^ *B*	412.29	1	412.29	44.37	<0.0002
*A* ^2^ *C*	473.48	1	473.48	50.96	<0.0001
*A* ^2^ *B* ^2^	424.89	1	424.89	45.73	0.0001
Residual	74.33	8	9.29		
Lack of Fit	17.80	3	5.93	0.52	0.68 (not significant)
Pure Error	56.54	5	11.31		
Cor Total	9395.63	19			

**Table 4 membranes-11-00956-t004:** Analysis of variance of the developed water flux based on the modified cubic model.

Source	Sum of Squares	Df	Mean Square	F-Value	*p*-Value
Model	26,605.60	10	2660.56	5.71	0.0075 (significant)
A-Feed concentration	3872.00	1	3872.00	8.31	0.0181
B-Pressure	4608.00	1	4608.00	9.90	0.0118
C-pH	313.60	1	313.60	0.67	0.4331
*AB*	128.00	1	128.00	0.27	0.6128
*BC*	128.00	1	128.00	0.27	0.6128
*A* ^2^	96.80	1	96.80	0.21	0.6592
*B* ^2^	288.80	1	288.80	0.62	0.4512
*A* ^2^ *B*	2073.60	1	2073.60	4.45	0.0641
Residual	4191.20	9	465.69		
Lack of Fit	1801.87	4	450.47	0.94	0.5091 (not significant)
Pure Error	2389.33	5	477.87		
Cor Total	30,796.80	19			

**Table 5 membranes-11-00956-t005:** Oil rejection and water flux at the optimized condition and corresponding values.

Run	Feed Concentration, (Ppm)	Pressure, (Bar)	pH	Oil Rejection, (%)	Water Flux, (L/m^2^·h)	Desirability
$Y_{PV}$	10	1.5	8.97	98.86	152.00	0.997
$Y_{RV}$	10	1.5	8.97	98.72	151.30	
Deviation. (%)				0.14	0.46	

$Y_{PV}$ is the predicted value of oil rejection and water flux and $Y_{RV}$ is the revalidated value of oil rejection and water flux.

**Table 6 membranes-11-00956-t006:** Comparison of the ceramic-based membrane, optimization method, factors, and responses.

Ceramic Material/Process	Experimental Design Method	Factor	Response	Author/Year
Nanocomposite membrane UF	CCD	Pressure: 3.0 bar pH: 9.0 Feed concentration: 600 ppm	Water flux: 152 L/m^2^·h Oil rejection: 98.72%	This work
Alumina MF	BBD	Transmembrane pressure: 3 bar; Feed flow rate: 300 L/h; Temperature: (60 °C)	Permeate flux: NA COD: 67%	[41]
Phosphate/kaolinite MF	BBD	Kaolinite: 15% Sintering temperature: 1000 °C, Sintering time: 2 h	Turbidity: 98.99%; TOC: 69.39%; COD: 74.00%; BOD: 77.11%	[45]
NA/MF	BBD	Pressure: 14.5 Pa; Feed velocity: 0.179 m/s; Pore size: 0.59 µm	Flux: NA	[48]
Mullite UF	BBD	pH: 7.2; Feed concentration: 921 mg/L, Coagulant concentration: 207 mg/L	Water flux: 123.85 L/m^2^·h; Oil rejection: 97.31%	[46]
Fly ash MF	CCD	Feed concentration: 176.07 mg/L; Pressure: 345 kPa	Flux: 936 L/m^2^·h; Oil rejection: 97.0%	[44]
Natural zeolite MF	CCD	pH: 7.04, Feed concentration: 75.00 mg/L, HFCM dosage: 0.35 g	Permeability: and ammonia removal: 96.5%	[47]

## Data Availability

Not applicable.

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
