# Peer review of "Optimization of a High-Performance Poly(diallyl dimethylammonium chloride)-alumina-perfluorooctanoate Intercalated Ultrafiltration Membrane for Treating Emulsified Oily Wastewater via Response Surface Methodology Approach"

_membranes, 2021, doi:10.3390/membranes11120956_

Round 1

Reviewer 1 Report

The work focused on the optimization of a high-performance UF membrane for treating emulsified oily wastewater.

While the work is interesting, the significant changes need to be made in order to improve the manuscript. 

Please, see the comments below:

1. The introduction does not provide sufficient background and does not include all relevant references.

Indeed, the justification for using the UF process for treating oily wastewater is not well presented. The advantages of the UF process should be better highlighted.

Moreover, more recent studies focused on the use of ceramic UF membranes for the treatment of oily wastewater should be cited. See for example:  

https://doi.org/10.1016/j.seppur.2020.118259

https://doi.org/10.1016/j.matchemphys.2020.124186

In addition, the advantages of ceramic membranes should be better emphasized.

2. The obtained results should be compared with those available in the literature.

3. The quality of all equations should be improved.   4. The quality of Figure 4 should be improved.   5. Why the changes in the permeate flux during the process are not presented? The fouling phenomenon should be discussed in detail.   6. The manuscript is not carefully written. Indeed, the information on Author Contributions, Funding, Institutional Review Board Statement, Data Availability Statement and Conflicts of Interest must be written. In turn, lines 549-558 are unnecessary.  

Reviewer 2 Report

Raji et al. presented the emulsified oily wastewater treatment using surface-functionalized ceramic UF membrane. The surface resonance method was used to evaluate the effect of feed concentration, pH, and applied pressure in a crossflow filtration setup on the membranes' water flux and oil rejection efficiency. For the oil rejection and water flux models, good correlation coefficients of 0.997 and 0.863 were obtained. The optimum feed concentration, pH, and applied pressure conditions were 10 ppm, pH 8.97, and 1.5 bar, respectively. Water flux and oil rejection were found to be at 152 LMH and 98.72%, respectively. The experimental values were also closely matching at these conditions. Overall, the work presented in the manuscript is interesting. The structure of the manuscript is logical. The manuscript can be accepted for publication after the authors address the following comments:

  1. Introduction section: Include a paragraph discussing various literature reports on the application of RSM for oily wastewater treatment.
  2. The no. of significant digits should be consistent while reporting the data in the manuscript.
  3. Properties of the oily wastewater, e.g., viscosity and density, used in this study should be reported.
  4. How did the authors prepare the feed solution?
  5. What was the oil droplet size distribution of the oily wastewater feed solution?
  6. What was the stability of the oily wastewater feed solution? Supplement with the necessary data.
  7. Consult a native speaker for correcting the English grammar and expression.

Round 2

Reviewer 1 Report

The manuscript has been corrected, thus, I recommend it for publication.

Reviewer 2 Report

The concerns have been addressed. The revised version is better. The manuscript can be accepted for publication.